# Effect of (NH_4_)_2_ZrF_6_, Voltage and Treating Time on Corrosion Resistance of Micro-Arc Oxidation Coatings Applied on ZK61M Magnesium Alloys

**DOI:** 10.3390/ma14237410

**Published:** 2021-12-03

**Authors:** Jiahui Yong, Hongzhan Li, Zhengxian Li, Yongnan Chen, Yifei Wang, Juanjuan Geng

**Affiliations:** 1School of Material Science and Engineering, Northeastern University, Shenyang 110819, China; yongjiahui423@163.com (J.Y.); lzxqy725@163.com (Z.L.); wyf86524@163.com (Y.W.); 2Northwest Institute for Nonferrous Metal Research, Xi’an 710016, China; gjjnin@163.com; 3Rare Mental Materials Surface Engineering Technology Research Center of Shaanxi Province, Xi’an 710016, China; 4School of Materials Science and Engineering, Chang’an University, Xi’an 710064, China; frank_cyn@163.com

**Keywords:** ZK61M alloy, micro-arc oxidation, corrosion property

## Abstract

The effects of (NH_4_)_2_ZrF_6_ concentration, voltage and treating time on the corrosion resistance of ZK61M magnesium alloy micro-arc oxidation coatings were studied by orthogonal experiments. The SEM result shows that the surface roughness and porosity of MAO coatings increased with (NH_4_)_2_ZrF_6_ concentration, voltage and treating time as a whole, except the porosity decreased with treating time. EDS, XRD and XPS analysis show that (NH_4_)_2_ZrF_6_ was successfully incorporated into coatings by reactive incorporation, coatings are dominantly composed of ZrO_2_, MgO, MgF_2_ and amorphous phase Mg phosphate. Potentiodynamic polarization was used to evaluate the corrosion property of coatings. When the concentration of (NH_4_)_2_ZrF_6_ is 6 g/L, the voltage is 450 V, and the treating time is 15 min, the coating exhibits the best corrosion resistance which corrosion current density is four magnitudes lower than substrate attributed to the incorporation of ZrO_2_ and the deposition of MgF_2_ in the micropores.

## 1. Introduction

Magnesium alloys are the lightest structural metallic materials, which are widely used in aerospace, automotive, 3C electronic products, biomedical applications due to their high specific strength, high specific stiffness, excellent shock absorption performance and electromagnetic shielding performance, good machinability, biocompatibility [1,2]. However, the wider application of magnesium alloys is limited by its active chemical properties [3]. As a result, magnesium alloys often require a surface treatment that can protect magnesium alloys from corrosive media [4].

Micro-arc oxidation (MAO) is a surface modification method developed from conventional anodic oxidation; it can generate a well-adhered oxide ceramic coating on the surface of magnesium alloy; thus, the magnesium substrate can be effectively protected. And it is widely used to improve the performance of light metals such as Al, Mg, Ti and their alloys now [5,6]. In the MAO process, the substrate that as an anode is put into the electrolyte which we prepared, then apply a high voltage, there will be plasma discharge and then the surface of the sample converts the metallic substrate into an oxide coating [7]. It is well known that the MAO tratement is very complex, which involves chemical, electrochemical, thermochemical, plasma chemical, metallurgical processes [8,9].

Although, plenty of factors can affect MAO coating performance, such as electrolyte, the composition of electrolyte has obtained the widespread attention of many researchers [10]. As the main phase composition of the zirconium salt electrolyte MAO coatings, ZrO_2_ has such characteristics as high hardness, low thermal conductivity, good thermal and chemical stability; therefore, it is suitable to be introduced into MAO coatings as an effective phase composition for Mg alloys to enhance corrosion resistance [11,12,13]. Mashtalyar [14] investigated the influence of the ZrO_2_/SiO_2_ nanoparticles mixture on corrosion property of MAO coatings, and the conclusions confirmed that the incorporation of ZrO_2_/SiO_2_ nanoparticles can significantly improve the electrochemical characteristics of the composite coatings due to their high chemical stability.

Voltage, current density, treating time and duty cycle also play an essential role in the progress of coating growth and determining the structure [15,16,17,18]. Xu [17] studied how applied voltage affects the corrosion property; it was shown that coatings have the smallest pore size, a relatively low porosity, fewer defects and a relatively high thickness, leading to the best corrosion resistance at 350 V. Ur Rehman [19] indicated that the size of pancakes increases over processing time, and longtime processing could cause severe inner layer damages and result in poor corrosion parameters.

Although a majority of researchers analyzed the influences of zirconium salt concentration, voltage, or treating time on coating performances, they act as a single key element usually. While understanding their mutual influence will be more helpful for us to choose the optimal experimental parameters. This research aims to study the effects of (NH_4_)_2_ZrF_6_ concentration, voltage and treating time on the microstructure and corrosion resistance of ZK61M magnesium alloys MAO coatings through orthogonal experiment.

## 2. Materials and Methods

### 2.1. Samples and Coating Preparation Procedures

ZK61M magnesium alloys (Zn 5.0–6.0 wt.%, Zr 0.30–0.90 wt.%, Mn ≤ 0.10 wt.%, Al ≤ 0.05 wt.%, Si ≤ 0.05 wt.%, Cu ≤ 0.05 wt.%, Ni ≤ 0.05 wt.%, Fe ≤ 0.05 wt.%, Be ≤ 0.01 wt.% and Mg balance) were used in this research and cut into the size of 15 × 15 × 5 mm. Then a hole with a diameter of 2 mm was punched in one of the corners of the sample, which was used for connecting the wire. Before MAO processing, the surface of samples was ground by 800, 1000, 1200, 1500, 2000 grit waterproof abrasive paper, subsequently, ultrasonically cleaned with ethanol and deionized water, and then dried.

### 2.2. Experiment Process

The MAO treatment was carried out in T-25Z MAO equipment (Northwest Institute for Nonferrous Metal Research, Xi’an, China) with a frequency of 500 Hz, a duty cycle of 15%. The stainless steel plate as the cathode, the samples as the anode. NaH_2_PO_4_, NaF, (NH_4_)_2_ZrF_6_ were the main component of basic electrolyte, and (NH_4_)_2_ZrF_6_ was chosen as one of orthogonal factors. Orthogonal table L_9_(3^4^) was applied for experimental design, Table 1 was complete factors and levels of the orthogonal experiment. After the MAO process, the MAO coatings were washed using deionized water. Significantly, a bubbler should be used to prevent particle agglomeration in the electrolyte, and place the breaker in deionized water containing ice to prevent the electrolyte temperature from becoming too high.

### 2.3. Characterization

Surface and cross-sectional morphologies of the coatings were examined by scanning electron microscopy VEGA II XMU (OXFORD, Oxford, UK), and the element distribution on the surface and cross-section were analyzed with an energy dispersive spectrometer equipped by SEM. The phase composition of coatings was performed by D/max 2200 PC X-ray diffractometer (RIGAKU, Tokyo, Japan) with Cu Kα radiation at 45 kV and 200 mA, and the detected range was between 20–80° with a step size of 0.02° and a scanning rate of 2°/min. Atom force microscopy (AFM, Bruker, Massachusetts, MA, USA) was applied to observe a three-dimension pattern and obtain surface roughness of coatings. Surface porosity of the MAO coatings was measured by Image J software, and the thinkness of each MAO coating was measured by thickness gauge five times, and then take the average value. Chemical states of the coating elements were analyzed by Dimension Icon X-ray photoelectron spectroscopy (Bruker, Massachusetts, MA, USA).

Electrochemical such as open circuit potential (OCP), potentiodynamic polarization (PDP) were examined using an electrochemical workstation VersaSTAT 3F (AMETEK, Pennsylvania, PA, USA) to evaluate the corrosion behavior of the MAO coatings in 3.5 wt.% NaCl solution at room temperature. The electrochemical experiment used a typical three-electrode system: a saturated calomel electrode (SCE) as reference electrode, the MAO coated sample as working electrode (1 cm^2^ exposed area), a Pt grid as a counter electrode. The polarization curves were measured at a scanning rate of 2 mV/s, and the potential range of −0.5 V to 2.0 V. The corrosion parameters such as anodic slopes (*b_a_*), catholic slopes (−*b*_c_) were extracted by Tafel extrapolation method. Polarization resistance (*R_p_*) was calculated by Stern-Geary Equation (1) which represents the corrosion property of MAO coatings.
(1)$$\begin{array}{c} R_{p}=\frac{b_{a}\cdot \left|b_{c}\right|}{2.303\cdot I_{corr}\cdot \left(b_{a}+\left|b_{c}\right|\right)} \end{array}$$

## 3. Results

### 3.1. Orthogonal Experimental Results and Analysis

Table 2 shows the results of orthogonal experiment. Corrosion current density (*I_corr_*), surface roughness (*Ra*) abstracted from AFM (Figure 1) and porosity were used as measures of coating quality, the corresponding samples were named 1^#^, 2^#^, 3^#^, 4^#^, 5^#^, 6^#^, 7^#^, 8^#^, 9^#^. k_1_ and k_2_ represent the mean values of the corresponding level of each factor, R stands for extremely poor, and its value is proportional to the importance of the coatings. As shown in Table 2, the influences of each factor on corrosion current density are: treating time > voltage > (NH_4_)_2_ZrF_6_ concentration, and the best scheme is A_3_B_3_C_3_; As for roughness, the importance is: (NH_4_)_2_ZrF_6_ concentration > voltage > treating time, the best scheme is: A_2_B_2_C_3_; As for porosity, the importance is: (NH_4_)_2_ZrF_6_ concentration > voltage > treating time, the best scheme is A_1_B_1_C_3_. Above all, the result of the three schemes is totally different. However, what we need to understand is that our ultimate goal is to judge the corrosion resistance, surface roughness and porosity are mostly like an auxiliary basis for evaluating corrosion resistance. So the treating time is the first important factor, followed by the (NH_4_)_2_ZrF_6_ concentration and voltage.

According to Table 2, when treating time increases, *I_corr_* first increase and then decrease, showing an overall downward trend; the roughness first decrease and then increase, showing an overall upward trend; the porosity has a downward trend as treating time increase. Therefore, the treating time should be 15 min. In the same way, when the concentration (NH_4_)_2_ZrF_6_ changes from 3 g/L to 9 g/L, *I_corr_* keeps decreasing, and the porosity keeps increasing; the roughness first decreases and then increases, showing an overall upward trend. So the (NH_4_)_2_ZrF_6_ concentration should be 6 g/L.

As the voltage increase, *I_corr_* first increases and then decrease, showing an overall downward trend; the roughness first decreases and then increase, showing an overall upward trend; the porosity has always been an upward trend. Hence, the voltage should be 450 V. And we can conclude that A_2_B_2_C_3_ is the best parameter combination for the comprehensive property of the coatings.

### 3.2. Surface Morphologies and Elemental Composition of MAO Coatings

Figure 2 presents the surface morphologies of the MAO coatings formed based on the orthogonal experiment. As shown in Figure 2, the surface appearances showed a poor coating uniformity and exhibited a typical irregular volcano-like morphology caused by local microdischarge events and gas evolution which occurred in the micro-arc oxidation stage and accompanied by microcracks caused by rapid solidification of molten oxides in cool electrolytes [20,21,22].

With the change of concentration, voltage and treating time, the coating morphologies have been to some extent modified and the changing of surface roughness and porosity can be observed from Figure 3. According to Figure 2 and Figure 3, the MAO coatings which with the content of 6 g/L (NH_4_)_2_ZrF_6_ have the lowest roughness. In addition, the porosity keeps increasing with the increase of (NH_4_)_2_ZrF_6_ and reaching the maximum at 9 g/L. This may be because there was strong spark discharge at low voltage due to the excessively high concentration of zirconium, which increase surface roughness and porosity.

Furthermore, the SEM images of Figure 2 show when the voltage is 500 V, there are more microcracks and large micropores on the surface of MAO coatings compared with other coatings obtained in 400 V and 450 V, which has high porosity and surface roughness, leading to poor surface compactness. The increase in porosity can arise from the increased plasma discharge energy due to the rise of voltage. During the MAO process, a low surface porosity can obtained by applied low voltage. Because low voltage can only produce a low discharge intensity, so the size of the formed discharge channel was smaller [17]. When the voltage was increased, the discharge energy was larger, too. Causing more heat to be generated during the MAO tratement, and then there would be more oxygen and molten products, it will induce serious packing behaviors of molten materials on the surface and result in larger discharge pores and high surface roughness [23].

Figure 2a,f,h are the surface morphologies with a treating time of 5 min, Figure 1b,d,i are the surface morphologies with a treating time of 10 min, Figure 2c,e,g are the surface morphologies with a treating time of 15 min. When the treating time is 10 min, they have the smallest surface roughness, and the porosity decreased with treating time prolonged, they have the lowest porosity when the treating time is 15 min. This is because in the early time of MAO process, coatings can be breakdown easily due to their low thinness, and the discharge is uniform, which produced uniform coatings. When the treating time is extended, the coatings continue to grow and thicken, which makes it difficult to breakdown, in addition, the number of sparks is reduced and inhomogeneous distributed, so the discharge energy of a single spark increases, resulting in a rough surface with more big micropores. Moreover, we can also see that there are more micropores were filled by sediment, which makes the porosity decrease.

During the MAO process, it is reported that particles will move to the anode under a strong electric field, and react with other ions in the electrolyte or the molten substrate, thereby being incorporated into the coating or deposited in the discharge channel [24]. According to the SEM images, we can see part of micropores were filled by sediments. In order to figure out the composition of the filler in the microcpores, we did a point scan by EDS at point A, B, C, D and E. The result is shown in Figure 4, the element composition is mainly F, Mg, O, P, presumably MgF_2_ and phosphate.

In the meantime, in order to figure out the relative contents of the main elements of MAO coating surfaces, EDS surface scan was used. And Table 3 shows the result of elemental composition of the coating surfaces. It demonstrates that the coatings are composed of Zr, Mg, O, F predominantly, and also contains Na, P and Zn. We can see the content of zirconium increase first and then decrease with the increase of (NH_4_)_2_ZrF_6_ concentration and treating time, and it keeps increasing when the voltage increase from 400 V to 500 V. The content of F has also changed significantly, increased with the increasing of (NH_4_)_2_ZrF_6_ concentration and treating time. According to the literature [22,25], the uniformity of MAO coatings can be improved by F^-^ effectively. But as shown in Figure 4, in this experiment, conducive to improving the corrosion resistance of MAO coating maybe its other function.

### 3.3. Cross-Sectional Morphologies and Elemental Composition of MAO Coatings

Figure 5 shows the cross-sectional images of MAO coatings formed in different parameters. It clearly shows a two-layered structure was formed on coatings, which had an outer porous layer appearing to be thicker with bigger pores and microcracks than the dense inner barrier layer. There was no obvious cracking at the interface between coating and substrate, indicating that they are well bonded.

Generally, the greater the applied voltage, the grater the thickness and porosity of the micro-arc coatings [26]. Observing the structure of the MAO coatings Figure 5a–c in more detail, as the voltage increases, the coatings were thicker, but there were more microspores, and we can observe a penetrating crack from Figure 5c which may be because of the fierce reaction. This would accelerate the failure of coating, but it can be reduced by the increase of (NH_4_)_2_ZrF_6_ concentration.

From Figure 5a,d,g, it is clearly seen that increasing the concentration of (NH_4_)_2_ZrF_6_, the number and size of micropores were reduced, and when the concentration was 6 g/L, coatings were the thickest. The comparison of the Figure 5c,e,g indicate that the micropores and microcracks on the outer porous layer became substantially smaller in size and number when the treating time was 15 min.

To further confirm the incorporation of particles into MAO coatings, EDS mapping analysis was performed to identify the particle distribution (Figure 6). According to Figure 6, the distribution of the element Mg and F were evenly distributed and similar in the coatings, presumably it may be MgF_2_. The uniform distribution of Zr in the coating proves that (NH_4_)_2_ZrF_6_ is reactively incorporated into the coating and may form ZrO_2_. As we all know, ZrO_2_ particle has good corrosion resistance, high hardness and it will undergo a phase change during the micro-arc oxidation process. ZrO_2_ exists as a monoclinic form (*m*-ZrO_2_) at room temperature, it will transform into a tetragonal (*t*-ZrO_2_) accompanied by a volume shrinkage (5%) when the temperature rises to 1100–1200 °C, and t-ZrO_2_ will transform into *m*-ZrO_2_ while cooling which is accompanied by volume expansion. The volume expansion and shear stress caused by the transition from *c*-ZrO_2_ to *m*-ZrO_2_ will reduce the pore diameter and inhibit or reduce the generation and propagation of microcracks.

### 3.4. Phase and Chemical Composition of MAO Coatings

The XRD patterns of the MAO coatings prepared in different conditions are displayed in Figure 7. The phase constituents of nine coatings are mainly composed of ZrO_2_, MgF_2_ and MgO. We speculate that (NH_4_)_2_ZrF_6_, NaF were successfully incorporated into the coatings due to the diffraction peak of MgF_2_ and ZrO_2_ were detected. The strong Mg peaks originating from the magnesium alloy substrate were detected because the coatings were too thin and X-ray could reach the substrate through the coatings. However, the peaks of phosphate compounds (e.g., Mg(PO_3_)_2_, Mg_3_(PO_4_)_2_ could not be found by XRD, which indicated that the amorphous phase of Mg phosphate may be produced in coatings [27].

Furthermore, in order to confirm the reaction products and the chemical nature of elements constituting of MAO coatings, we applied XPS analysis, and the result of XPS survey spectra of coatings are presented in Figure 8. It is shown that the coating consists of Mg, Na, F, O, Zr and P, which confirmed the incorporation of (NH_4_)_2_ZrF_6_ into the surface layers. Figure 8 shows the specific binding energies of Zr, Mg, O, F elements of MAO coatings. Table 4 show the bindng energy of main elements and corresponding compounds. We can see that at 185.5 eV, 183.3 eV are the peaks of Zr 3d_3/2_ and Zr 3d_3/5_, which represent ZrO_2_. The Mg 1s spectrum can be divided into two peaks: Mg 1s peak at 1303.9 eV is corresponding to MgO, and the other peak at 1306.5 eV is from MgF_2_. The O 1s spectrum can be deconvoluted into two peaks at binding energies of 531.3 eV, 532.1 eV [28], respectively. The peak located at the binding energy of 531.3 eV, revealing the existence of ZrO_2_ [29]. The peak at the binding energy of 532.1 eV, is attributed to the MgO. The peak located at 685.3 eV and 686.1 eV correspond to MgF_2_. This is consistent with the result of XRD.

### 3.5. Electrochemical Corrosion Test of MAO Coatings

The electrochemical experiment was carried in 3.5% NaCl solution at 25 °C, and Figure 9 displays the polarization curve of the MAO coatings and Mg substrate. Table 5 show the corrosion potential (*E_corr_*), corrosion current density (*I_corr_*), anodic Tafel slope *(b_a_*), cathodic Tafel slope (*b_c_*) which extracted from the potentiodynamic polarization plots via Tafel region extrapolation and polarization resistance (*R_p_*) calculated using the Stern-Geary equation. There are obvious differences in the polarization curves of MAO coatings under different conditions. Compared to the uncoated samples, the coated samples exhibit more positive corrosion potential, lower corrosion current density, these indicate better corrosion resistance they perform [30,31]. It can be seen from Table 5 that NO.5^#^ has the smallest *I_corr_* of 3.247 × 10^−8^, which is foue orders of magnitude lower than the uncoated sample (1.526 × 10^−4^). And the *E_corr_* is increased from −1.594 V to −1.459 V. Moreover, the polarization resiatance of NO.5^#^ is also the largest among all samples. All this prove the NO.5^#^ has the best corrosion properties of coatings which were obtained in the parameters with 6 g/L (NH_4_)_2_ZrF_6_, 450 V voltage and 15 min treating time.

Many factors such as microstructure, surface roughness, porosity, element composition and coating thickness can affect the corrosion perporty of coatings. The corrosion property of coatings increased as thickness increased, as shown in Figure 4 and Table 4. Micropores and microcracks are the weakness of coatings, and these parts are relatively less resistant to Cl^−^ [11]. This is why the NO.2^#^ shows the worst corrosion resistance among all the coatings. Moreover, the low content of ZrO_2_ may be the other reason why the sample suffers such serious corrosion failures.

Besides, it is interesting to notice that No.5^#^ has the best corrosion resistance even though it has high porosity (15.196%). The high corrosion property is due to the increase of ZrO_2_, which has higher chemical resistance than other phase compositions in the coatings. This can also be proved in Lee’s research [32], which clarified that the incorporation of ZrO_2_ can improved corrosion property of MAO coatings effectively. It should be noted that there exist lots of MgF_2_ concentrated in the micropores, according to Liu’s research [33], the deterioration rate of the MAO coating compounds can be reduced by the presence of MgF_2_ when MAO coating was exposed to corrosive medium, so the existence of MgF_2_ is of great help in preventing corrosive ions from reaching substrate surface quickly [34]. In addition, this is also the result of compacter structures and thicker coatings working together. And NO.5^#^ has the largest polarization resistance, which also proves this.

From Figure 9, we can see that all coatings exist the breakdown potential (*E_bd_*) on polarization curves. At *E_bd_*, the *I_corr_* on the coatings change greatly, indicating that coating breakdown occurs. Moreover, the I_corr_ of coatings soars after *E_bd_*, which can be attributed to the severe pitting corrosion after coating breakdown. And, galvanic corrosion maybe also contributes to the dramatic increase of *I_corr_* because of the more positive *E_corr_* of the coatings with respect to *E_corr_* of the substrate [35].

## 4. Conclusions

ZK61M magnesium MAO coatings were prepared by orthogonal experiment with the factors (NH_4_)_2_ZrF_6_ concentration, voltage and treating time, the microstructure, phase composition and corrosion behavior of coatings were studied. The following conclusions can be drawn:

The best coating was obtained from the combination of A_2_B_2_C_3_, which means the parameter combination is 6 g/L (NH_4_)_2_ZrF_6_, voltage is 450 V, treating time is 15min.The MAO coatings on ZK61M magnesium alloy mainly consisted of ZrO_2_, MgO, MgF_2_ and amorphous phase Mg phosphate.NO.5# has the best corrosion property which corrosion current density is 4 magnitudes lower compared to the substrate. This may attribute to the higher content of zirconium, compacter structure, thicker coating, and the filling effect of MgF_2_ in the micropores.

## Figures and Tables

**Figure 1 materials-14-07410-f001:**
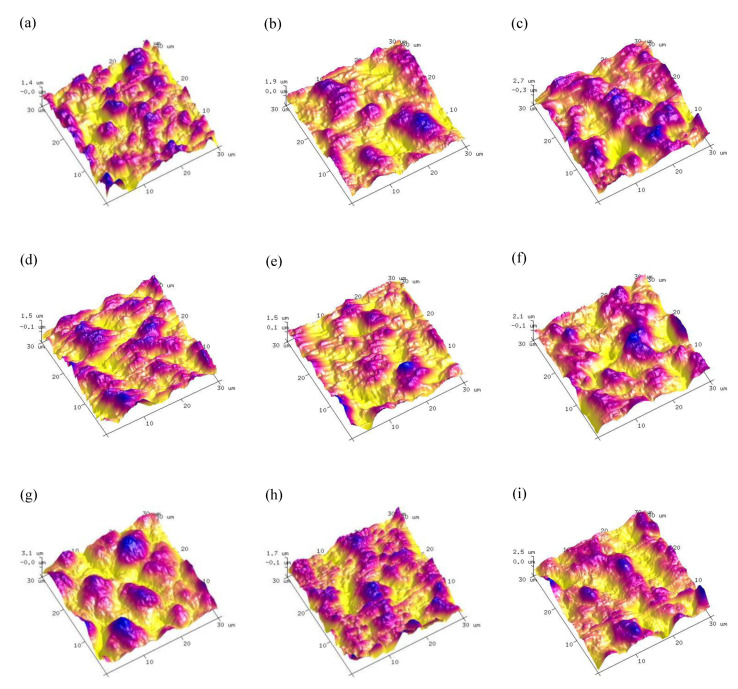
AFM image of MAO coatings obtained based on orthogonal experiment: (**a**–**i**) are 1^#^–9^#^ respectively.

**Figure 2 materials-14-07410-f002:**
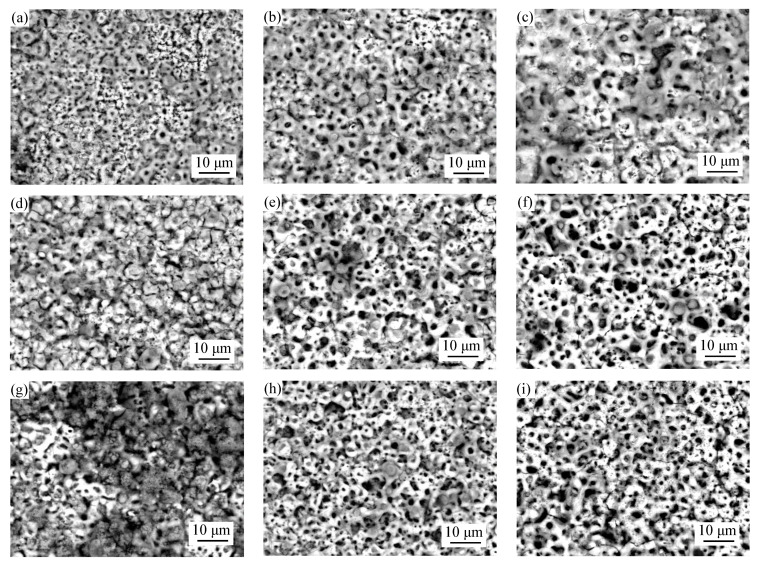
Surface morphology of coatings obtained based on orthogonal experiment: (**a**–**i**) are 1^#^–9^#^ respectively.

**Figure 3 materials-14-07410-f003:**
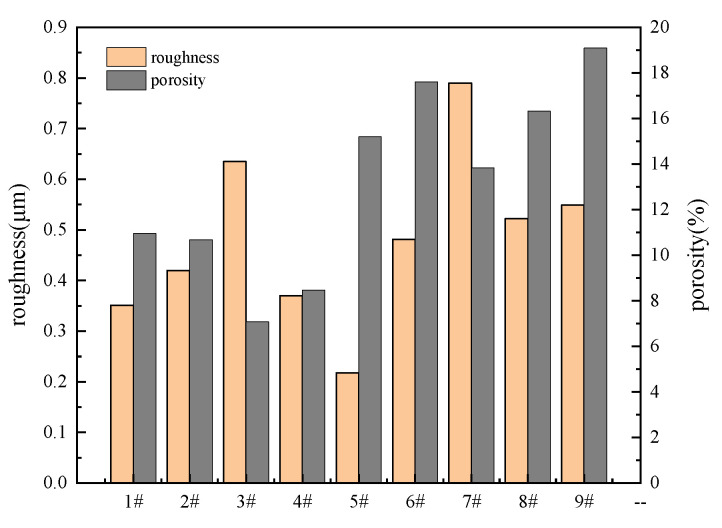
Variation of roughness and porosity of MAO coatings formed with different parameters.

**Figure 4 materials-14-07410-f004:**
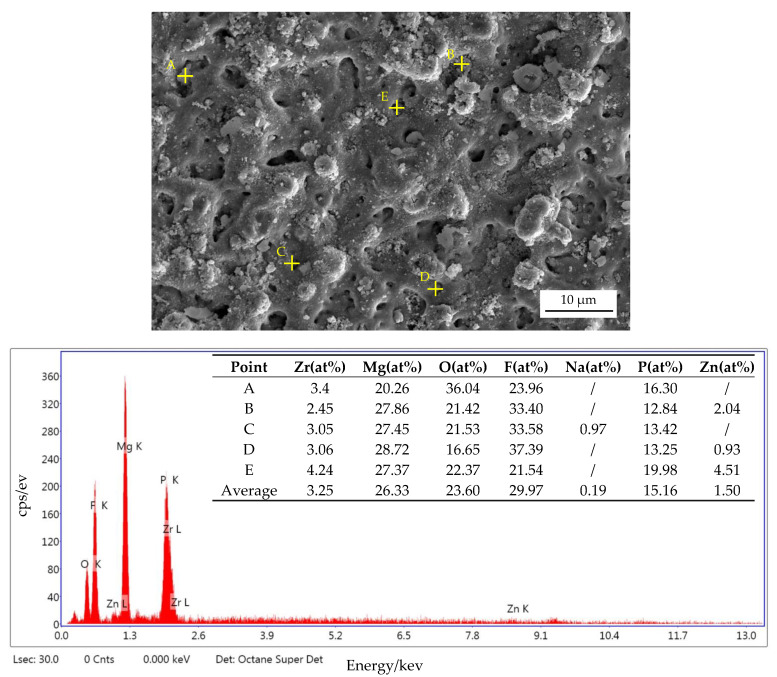
Surface morphology and EDS analysis of NO.5^#^.

**Figure 5 materials-14-07410-f005:**
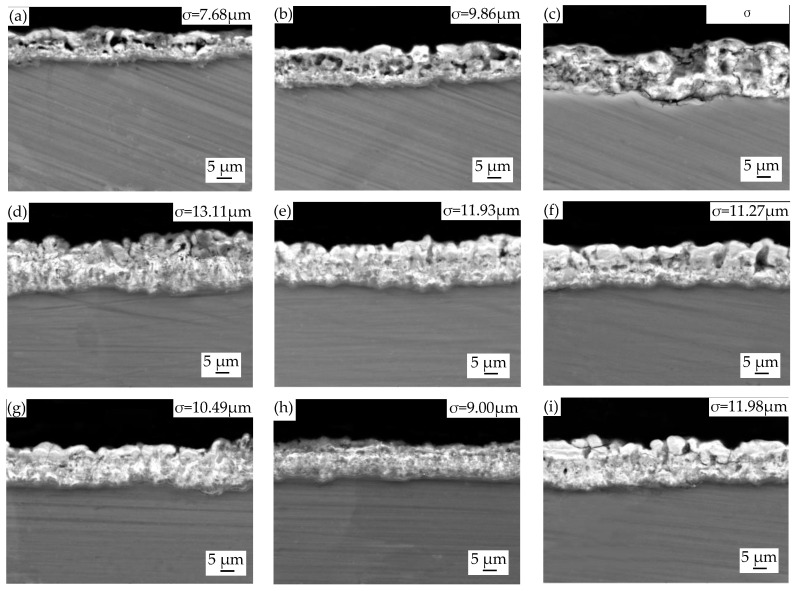
SEM images of cross-sectional morphology of coatings formed in different parameters: (**a**–**i**) are 1^#^–9^#^, respectively.

**Figure 6 materials-14-07410-f006:**
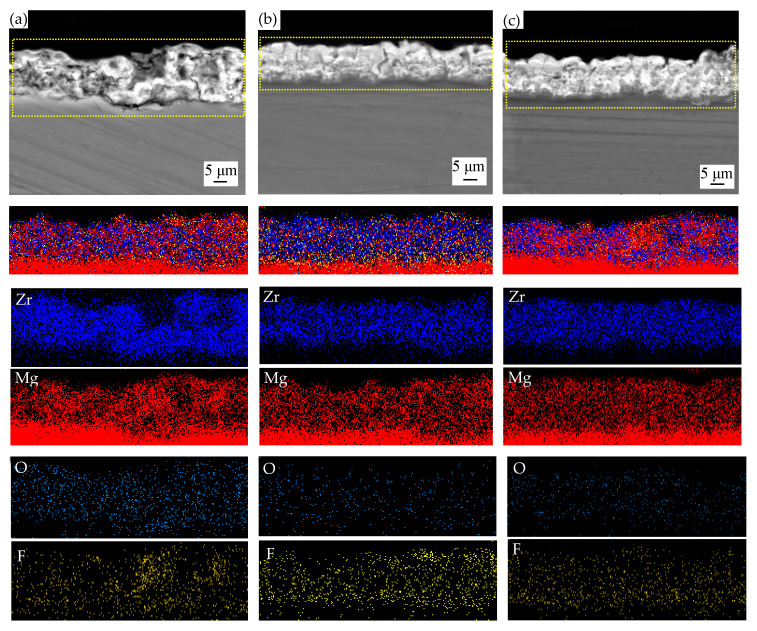
The cross-sectional morphologies and EDS mapping of the coatings: (**a**) NO.3^#^; (**b**) NO.5^#^; (**c**) NO.7^#^.

**Figure 7 materials-14-07410-f007:**
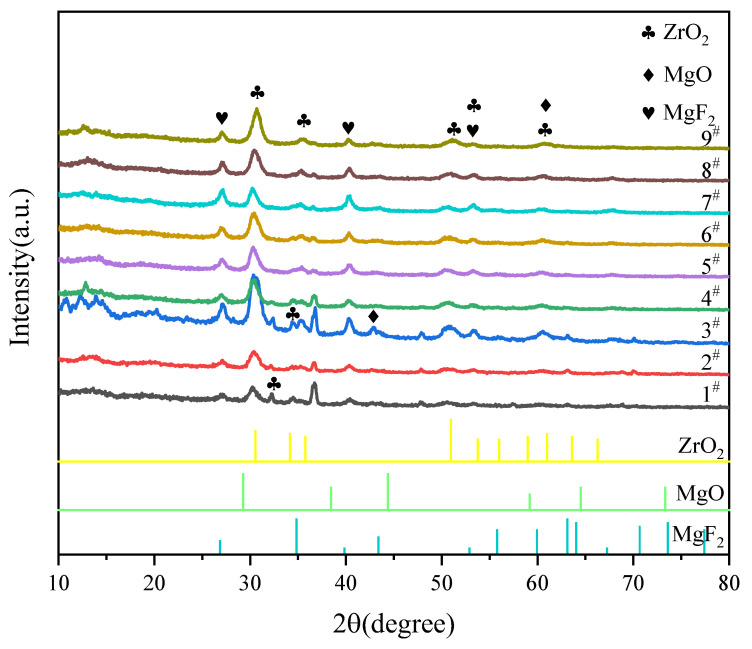
XRD patterns of MAO coatings from orthogonal experiment.

**Figure 8 materials-14-07410-f008:**
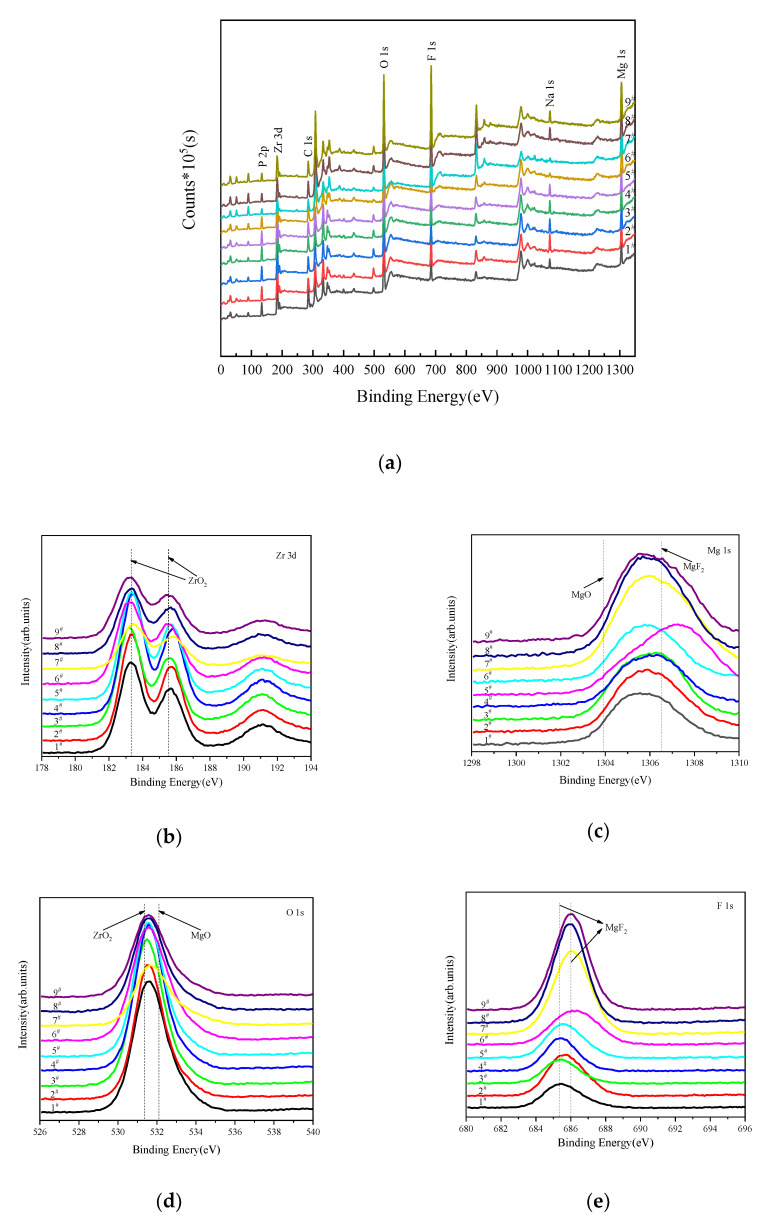
XPS analysis of MAO coatings formed in orthogonal experiment: (**a**) XPS survery spectra, (**b**) Zr 3d; (**c**) Mg 1s; (**d**) O 1s; (**e**) F 1s.

**Figure 9 materials-14-07410-f009:**
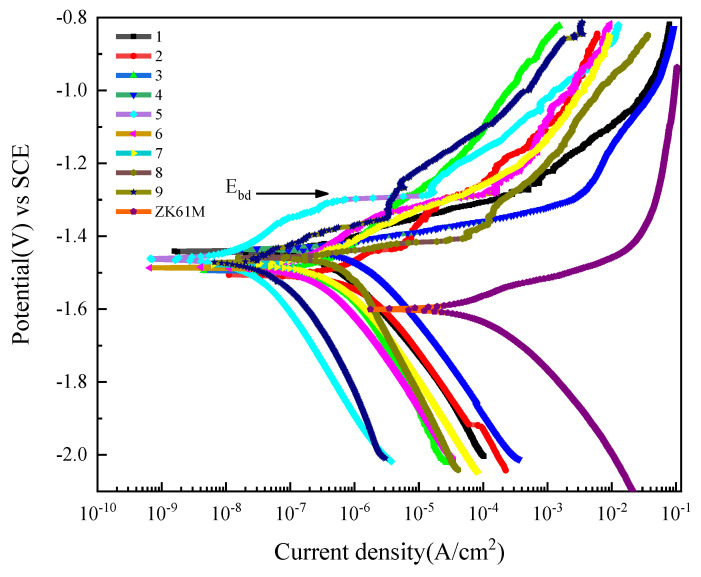
Potentiodynamic polarization curves of ZK61M magnesium alloy and MAO coatings (1^#^–9^#^) in 3.5 wt.% NaCl solution at room temperature.

**Table 1 materials-14-07410-t001:** Factors and levels of the orthogonal experiment.

Factors	A	B	C
Levels	(NH_4_)_2_ZrF_6_ Concentration (g/L)	Voltage (V)	Treating Time (min)
1	3	400	5
2	6	450	10
3	9	500	15

**Table 2 materials-14-07410-t002:** Orthogonal experimental array and experimental results.

Experimental	A	B	C	*I_corr_* (mA/cm^2^)	*R_a_* (μm)	Porosity (%)
1^#^	3	400	5	0.2604	0.351	10.957
2^#^	3	450	10	1.1740	0.420	10.685
3^#^	3	500	15	0.6487	0.635	7.076
4^#^	6	400	10	1.025	0.370	8.471
5^#^	6	450	15	0.03247	0.218	15.196
6^#^	6	500	5	0.3394	0.481	17.602
7^#^	9	400	15	0.4722	0.790	13.833
8^#^	9	450	5	0.7529	0.522	16.323
9^#^	9	500	10	0.05712	0.549	10.09
k1	0.694 (0.469) (9.573)	0.586 (0.504) (11.087)	0.451 (0.451) (14.961)			
k2	0.466 (0.356) (13.756)	0.653 (0.387) (14.068)	0.752 (0.446) (12.749)			
k3	0.427 (0.620) (16.415)	0.348 (0.555) (14.589)	0.384 (0.548) (12.035)			
R	0.267 (0.264) (6.842)	0.305 (0.168) (3.502)	0.368 (0.102) (2.926)			

**Table 3 materials-14-07410-t003:** EDS analysis of the coating surface (1^#^–9^#^) prepared in different parameters.

Coating	Zr (at%)	Mg (at%)	O (at%)	F (at%)	Na (at%)	P (at%)	Zn (at%)
1^#^	13.27	30.43	23.91	15.81	0.87	14.90	0.81
2^#^	11.27	32.07	21.88	18.61	1.00	13.96	1.22
3^#^	11.54	32.54	19.48	23.41	0.98	11.02	1.03
4^#^	15.72	26.41	18.50	28.44	1.35	8.53	1.05
5^#^	16.26	27.36	20.16	24.07	1.15	10.10	0.90
6^#^	15.87	29.18	19.86	25.12	1.18	8.23	0.57
7^#^	13.48	30.61	14.11	33.58	0.88	6.43	0.91
8^#^	14.07	29.22	16.49	32.57	0.91	5.88	0.87
9^#^	19.86	26.72	18.67	28.02	1.08	4.93	0.73

**Table 4 materials-14-07410-t004:** The binding energy of the main elements and corresponding compounds for MAO coatings formed in orthogonal experiment.

Sample	Zr 3d	Mg 1s	O 1s	F 1s
Binding Energy (eV)	185.5	1306.5	531.3	685.3
Compounds	ZrO_2_	MgF_2_	ZrO_2_	MgF_2_
Binding Energy (eV)	183.3	1303.9	532.1	686.1
Compounds	ZrO_2_	MgO	MgO	MgF_2_

**Table 5 materials-14-07410-t005:** Parameter values of potentiodynamic polarization curves of ZK61M magnesium alloy and coatings in 3.5 wt.% NaCl solution at room temperature.

Coating	*E_corr_* (V)	*I_corr_* (A/cm^2^)	*b_a_* (mV/dec)	*−b_c_* (mV/dec)	*R_p_* (kΩ·cm^2^)
ZK61M	−1.594	1.526 × 10^−4^	87.29	176.68	1.662 × 10^2^
1^#^	−1.443	2.604 × 10^−7^	52.84	150.08	6.517 × 10^4^
2^#^	−1.498	1.174 × 10^−6^	123.58	238.52	3.010 × 10^4^
3^#^	−1.491	6.487 × 10^−7^	179.34	308.22	7.588 × 10^4^
4^#^	−1.433	1.025 × 10^−6^	37.92	211.78	1.362 × 10^4^
5^#^	−1.458	3.247 × 10^−8^	84.21	291.75	8.739 × 10^5^
6^#^	−1.481	3.394 × 10^−7^	94.97	263.97	8.935 × 10^4^
7^#^	−1.475	4.722 × 10^−7^	98.87	239.63	6.429 × 10^4^
8^#^	−1.465	7.529 × 10^−7^	44.00	329.77	2.239 × 10^4^
9^#^	−1.476	5.712 × 10^−8^	93.55	247.76	5.162 × 10^5^

## Data Availability

Data sharing Not applicable.

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
