# Peer review of "Effect of (NH4)2ZrF6, Voltage and Treating Time on Corrosion Resistance of Micro-Arc Oxidation Coatings Applied on ZK61M Magnesium Alloys"

_materials, 2021, doi:10.3390/ma14237410_

Round 1

Reviewer 1 Report

In this research the authors investigated the effects of 67 (NH4)2ZrF6 concentration, voltage and treating time on the microstructure and corrosion 68 resistance of ZK61M magnesium alloys MAO coatings through orthogonal experiment. Authors need to address several major issues to make the paper suitable for publication:

  1. provide the information how you measured the roughness, porosity, etc. in table 1. If AFM is used for roughness, provide the actual scanning picture.
  2. In figure 3, only one point's EDS has been presented, authors should analyse at least 5 points and make the average and present one EDS spectra for evidence. 
  3. Same applies for table 3, authors should analyse more points and provide the error bar. 
  4. In figure 4, the thickness of the coating should be indicated  within each picture for more clarity. 
  5. In figure 5, the authors should provide also the combined EDS mapping with all the elements in one image for all three samples for evidence and more clarity. 
  6. For the Phase and Chemical Composition of MAO Coatings, authors should provide all the XRD reference of standard PDF card. There are many small peaks which are not identified, all peaks should be designated. For the XPS analysis, the survey spectra for all the samples in one image should be provided. The XPS explanations were made without any reference. Authors should provide a table with the binding energy and corresponding all possible chemical compounds with ref. 
  7. Even though no plagiarism has been detected from a single source, software picked several matching from different sources, authors should paraphrase those properly. here are some list of pages which need action:  40-42; 107-109; 169-173; 188-190; 126-128; 147-150; etc. 

Author Response

Point 1: Provide the information how you measured the roughness, porosity, etc. in table 1. If AFM is used for roughness, provide the actual scanning picture.

Response 1: The roughness was measured by AFM, porosity was measured by Image J software. AFM scanning picture was shown in Figure 1.

Point 2: In figure 3, only one point's EDS has been presented, authors should analyse at least 5 points and make the average and present one EDS spectra for evidence.

Response 2: Average value of 5 points and EDS spectra were shown in Figure 4.

Point 3: Same applies for table 3, authors should analyse more points and provide the error bar.

Response 3: Table 3 is the result of EDS face scan. We think it is relatively accurate, so we did not analyze more areas.

 Point 4: In figure 4, the thickness of the coating should be indicated within each picture for more clarity.

Response 4: As shown in Figure 5, the coating thickness of each sample was marked in the cross-sectional image.

Point 5: In figure 5, the authors should provide also the combined EDS mapping with all the elements in one image for all three samples for evidence and more clarity.

Response 5: Figure 6 provides the image of the combined EDS mapping.

Point 6: For the Phase and Chemical Composition of MAO Coatings, authors should provide all the XRD reference of standard PDF card. There are many small peaks which are not identified, all peaks should be designated. For the XPS analysis, the survey spectra for all the samples in one image should be provided. The XPS explanations were made without any reference. Authors should provide a table with the binding energy and corresponding all possible chemical compounds with ref.

Response 6: As shown in Figure 7, all the XRD reference of standard PDF card was provided and the remaining small peaks were marked. XPS survey spectra for all samples in one image was shown in Figure 8, and the table containing binding energy, chemical compounds was shown in Table 4.

Point 7: Even though no plagiarism has been detected from a single source, software picked several matching from different sources, authors should paraphrase those properly. here are some list of pages which need action:  40-42; 107-109; 169-173; 188-190; 126-128; 147-150; etc.

Response 7: 40-42 cited from [7]. 107-109 were written by myself. To avoid unnecessary misunderstandings, I have made some changes. 126-128 (now is 129-132) was written by myself. 169-173 (now is 191-195) were summarized from [17] and [23]. 188-190 cited from [24]. 147-150

were summed up by me from [20-22], and me have made some adjustments.

Reviewer 2 Report

The present manuscript reports the effect of (NH4)2ZrF6 concentration, voltage and treating time on the corrosion resistance of ZK61M magnesium alloy. The experiments seem to be conducted systematically using appropriate characterization methods like SEM-EDS, XRD, XPS, OCP, PDP. The authors made conclusions on the best coating condition and the composition of MAO coating. Unfortunately, the discussion of the results presented is sketchy or not proper. There is also no proper literature discussion although there are several authors who investigated very similar research. After revisions including a proper assessment of relevant literature, more detailed discussions of the results, and some clarifications the manuscript can be published in Materials.

- Since the resolution of EDS has limited and the quantitative error analysis of EDS is a combination of precision and accuracy, point analysis for one point seems to be not enough.

- Some table including Chinese characters is overlapped in paragraphs.

- The roughness results of 1# and 5# look like almost same level. Please discuss or explain it.

Author Response

Point 1: Since the resolution of EDS has limited and the quantitative error analysis of EDS is a combination of precision and accuracy, point analysis for one point seems to be not enough.

Response 1: As shown in Figure 4, I took 5 points on the surface of NO.5# for EDS point scanning, and then took the average value.

Point 2: Some table including Chinese characters is overlapped in paragraphs.

Response 2: Sorry about that, I did not find the situation you mentioned, so I didn’t optimize it.

Point 3: The roughness results of 1# and 5# look like almost same level. Please discuss or explain it.

Response 3: Sorry about that, it is my mistake, the error has been fixed in Table 2. The previous date is the date I did not eliminate the experimental error, the scan picture is shown in (a). The date after eliminating the error can better reflect the true roughness of the coating, and the scan picture is shown in (b). I also included some literature discussions in the article.

Round 2

Reviewer 1 Report

Authors have resolved the issues and have improved the manuscript sufficiently. However, authors still need to edit and paraphrase several lines of the manuscript to reduce the matching with the other sources. I did not ticked on the plagiarism detected options as I think that it is very minor and authors can easily fix that. For any internationally reputed journal it is important to not have substantial matching/plagiarism. For authors convenience, I have attached the plagiarism report. instead of reference section, authors need to go through the whole manuscript and edit wherever needed. After that I will accept the manuscript.    

Author Response

Dear reviewer:

Thank you for providing me the plagiarism report. According to the content marked in the report, I have revised my manuscript and hope that it is now clear. Please see the revised part of the manuscript. But I still have three problems unresolved:

(1) Mark NO.4 is the content of the journal template. So I did not modify this part.

(2) At the end of the manuscript, such as “author contributions”, “funding”, “institutional review board statement”, “informed consent statement”, “date availability statement”, “acknowledgments”, “conflicts of interest” were also marked. I understand this part of the content as the fixed format of the journal, so I did not modify this part.

(3) All references in the manuscript were marked. And I checked the format of the references, unfortunately, I did not find the problem. For this reason, I chose not to make this change.

We would like to thank the referee again for taking the time to review our manuscript. If there are still inappropriate content modifications, please contact us. Looking forward to hearing from you.

Thank you and best regards.

Yours sincerely

Jiahui Yong

Hongzhan Li

[email protected]
